# Utilizing Feature Selection Techniques for AI-Driven Tumor Subtype Classification: Enhancing Precision in Cancer Diagnostics

**DOI:** 10.3390/biom15010081

**Published:** 2025-01-08

**Authors:** Jihan Wang, Zhengxiang Zhang, Yangyang Wang

**Affiliations:** 1Yan’an Medical College of Yan’an University, Yan’an 716000, China; 2School of Electronics and Information, Northwestern Polytechnical University, Xi’an 710129, China

**Keywords:** tumor subtype classification, feature selection, artificial intelligence, machine learning, deep learning, multi-omics, high-dimensional data, biomarkers

## Abstract

Cancer’s heterogeneity presents significant challenges in accurate diagnosis and effective treatment, including the complexity of identifying tumor subtypes and their diverse biological behaviors. This review examines how feature selection techniques address these challenges by improving the interpretability and performance of machine learning (ML) models in high-dimensional datasets. Feature selection methods—such as filter, wrapper, and embedded techniques—play a critical role in enhancing the precision of cancer diagnostics by identifying relevant biomarkers. The integration of multi-omics data and ML algorithms facilitates a more comprehensive understanding of tumor heterogeneity, advancing both diagnostics and personalized therapies. However, challenges such as ensuring data quality, mitigating overfitting, and addressing scalability remain critical limitations of these methods. Artificial intelligence (AI)-powered feature selection offers promising solutions to these issues by automating and refining the feature extraction process. This review highlights the transformative potential of these approaches while emphasizing future directions, including the incorporation of deep learning (DL) models and integrative multi-omics strategies for more robust and reproducible findings.

## 1. Introduction

Cancer remains one of the leading causes of morbidity and mortality worldwide, characterized by its remarkable complexity and heterogeneity [1,2,3]. Tumor heterogeneity manifests at various levels, including genetic, epigenetic, transcriptomic, and proteomic variations, which contribute to differing tumor behaviors, responses to treatment, and patient prognoses [4,5]. Understanding and accurately classifying tumor subtypes is critical for the advancement of personalized medicine, enabling tailored therapeutic strategies that can significantly improve patient outcomes [6,7].

The classification of tumors into distinct subtypes has evolved from a purely histopathological approach to a more integrated methodology that incorporates molecular and genomic information. The advent of high-throughput technologies, such as next-generation sequencing (NGS) and various omics approaches, has generated vast amounts of biological data [8,9], offering unprecedented insights into the molecular underpinnings of cancer. However, this wealth of data also presents a challenge: how to distill complex datasets into meaningful insights that facilitate effective tumor subtype classification [10]. Feature selection techniques emerge as essential tools in this context, serving to identify and prioritize the most informative variables from high-dimensional datasets [11,12]. Feature selection refers to the process of identifying the most informative variables from high-dimensional datasets to reduce dimensionality while retaining essential information. By reducing the dimensionality of data, feature selection not only enhances the interpretability of models but also improves the accuracy of classification algorithms [13,14,15]. This is particularly important in the field of oncology, where the identification of relevant biomarkers can lead to improved diagnostics, prognostics, and treatment strategies [16,17]. With the rise of artificial intelligence (AI), particularly machine learning (ML) and deep learning (DL) algorithms, the integration of AI-driven feature selection techniques has further enhanced the ability to manage and analyze large-scale datasets, thereby refining cancer subtype classification models [18,19]. ML is a branch of AI that involves training algorithms to recognize patterns and make predictions based on data. DL, a subset of ML, uses multi-layered neural networks to automatically learn hierarchical feature representations from raw data. These AI approaches have proven crucial in automating the extraction of relevant features, significantly improving diagnostic accuracy and personalized treatment planning.

In essence, feature selection acts as a bridge between raw data and actionable clinical insights, highlighting its indispensable role in cancer research. Feature selection techniques can be broadly categorized into three main types: filter methods, wrapper methods, and embedded methods [20,21,22]. Filter methods assess the relevance of features based on intrinsic properties of the data, often utilizing statistical tests to evaluate the relationship between features and the target variable [23]. Wrapper methods, on the other hand, involve evaluating the performance of a predictive model using different subsets of features, allowing for a more tailored selection process [24]. Embedded methods integrate feature selection into the model training process itself, identifying the most relevant features as part of model optimization [25]. Each of these approaches offers unique advantages and limitations, and the choice of technique can significantly influence classification outcomes [13,26].

The application of feature selection in tumor subtype classification has been demonstrated in various studies, revealing its potential to enhance the accuracy of predictions and inform clinical decision-making [27,28]. For instance, research has shown that specific gene expression profiles can serve as reliable indicators for distinguishing between subtypes of breast cancer, thereby guiding targeted therapy selection [29,30]. Similar advancements have been observed in other malignancies, such as lung cancer and colorectal cancer, where feature selection has played a pivotal role in improving diagnostic precision and therapeutic strategies [27,31,32,33]. However, the integration of feature selection techniques is not without its challenges. Data quality remains a pressing concern in cancer research, as inconsistencies, missing values, and variations in data generation methods can impact the reliability of feature selection outcomes [20,22,34]. Moreover, interindividual variability introduces additional complexity, as genetic and environmental factors can lead to diverse responses among patients with the same tumor subtype [13,35]. These challenges underscore the importance of adopting robust and standardized methodologies for feature selection to enhance the reproducibility and generalizability of findings [36]. Leveraging AI for data cleaning and preprocessing can address some of these challenges, allowing for more consistent feature extraction across diverse datasets. Furthermore, the risk of overfitting poses a significant concern when utilizing high-dimensional datasets [37,38]. Effective feature selection can reduce this risk by ensuring that only the most relevant features are retained for analysis, thereby improving the robustness of the model, which is critical for the effectiveness of subtype classification [4,39].

Previous reviews have addressed feature selection in various contexts. For example, one review highlights its role in dimensionality reduction and disease understanding through real-world medical case studies [40]. Another explores how feature selection mitigates the curse of dimensionality in disease risk prediction using genetic data [41]. A third survey examines feature selection alongside dimensionality reduction and classification techniques for chronic disease diagnosis, emphasizing their impact on predictive accuracy and computational efficiency [42]. Building upon these studies, our review focuses specifically on the role of feature selection techniques in cancer subtype classification. In this review, we provide a comprehensive exploration of feature selection techniques, their role in unraveling tumor complexity, and their contributions to improving subtype classification. We begin by highlighting the significance of accurately classifying tumors to advance personalized medicine, emphasizing the challenges posed by the heterogeneity of cancer. We then provide an overview of various feature selection methods, detailing their unique characteristics and applications in cancer research. The discussion on practical applications illustrates how these techniques enhance predictive accuracy and inform clinical decision-making across different tumor types. Additionally, we emphasize the transformative impact of AI integration in refining feature selection, demonstrating its potential in streamlining cancer diagnostics and advancing therapeutic strategies. Furthermore, we address the challenges and limitations inherent in feature selection, including data quality and the risks of overfitting. Finally, we outline future directions, emphasizing the need for integration of multi-omics data and advancements in computational methods.

## 2. Overview of Feature Selection Techniques

### 2.1. Fundamentals and Workflow of Feature Selection Techniques in High-Dimensional Data Analysis

Feature selection is a critical step in the data analysis process, especially in high-dimensional settings such as genomics, transcriptomics, and proteomics [38,39]. The goal of feature selection is to identify the most relevant variables from a dataset, thereby reducing dimensionality while preserving essential information. This not only enhances the performance of classification algorithms but also aids in interpreting the underlying biological significance of selected features [36,40,41]. For instance, one study presents a machine learning framework that integrates somatic mutation and gene expression data to classify breast cancer subtypes. The authors employ feature selection techniques to identify key genes associated with specific subtypes. These selected features improved model accuracy by reducing noise and overfitting while also providing biologically meaningful insights into the molecular mechanisms of breast cancer [43]. Similar approaches have been applied in other cancer types, emphasizing the versatility and effectiveness of feature selection in handling high-dimensional datasets and improving clinical decision-making. By focusing on the most informative features, these techniques enhance both the reliability and interpretability of predictive models, ultimately supporting advancements in precision medicine. Recent advancements in AI, particularly ML and DL, have further optimized feature selection techniques by automating the identification of relevant features, thereby improving the accuracy and interpretability of cancer subtype classification models. In general, the basic workflow of feature selection methods consists of four fundamental processes: generating feature subsets, evaluating feature subsets, determining stopping criteria, and validating results, as shown in Figure 1. In the subset generation phase, the search strategy is crucial; an effective search strategy should achieve optimal solutions, have local search capabilities, and ensure computational efficiency. The evaluation function guides the feature selection process by deciding whether a feature should be added to or removed from the feature subset. When the stopping criteria are met, the feature selection process concludes, and the selected feature subset is applied to the validation dataset for verification. AI-enhanced methods, such as reinforcement learning, have introduced new possibilities in optimizing the search strategy by learning from iterative feedback, enabling more precise feature subset generation and refinement.

### 2.2. Categorization of Feature Selection Techniques and Their Operational Mechanisms

In this review, we categorize feature selection into four primary approaches: filter methods [23,27], wrapper methods [37,44], embedded methods [25,45], and swarm intelligence methods [46,47], and each of these approaches has its unique advantages, limitations, and applications in cancer research, as shown in Figure 2. Here, we introduce swarm intelligence methods to reflect their increasing importance in modern applications [48]. Swarm intelligence algorithms are often viewed as an organic combination of the three conventional feature selection methods—filter, wrapper, and embedded techniques. These algorithms leverage the strengths of each approach by combining the statistical evaluation of features (from filter methods), the model performance optimization (from wrapper methods), and the integration of feature selection within the model training process (from embedded methods). The integration of AI has allowed for more dynamic, hybrid approaches that combine the strengths of different feature selection techniques. For example, the combination of wrapper methods with DL models can lead to more robust feature selection by identifying nonlinear patterns in complex biological data.

#### 2.2.1. Filter Methods

Filter methods operate independently of any ML algorithm and rely on statistical measures to evaluate the relevance of features. These methods assess the relationship between each feature and the target variable (e.g., tumor subtype) before any modeling occurs. Common filter techniques include the following:

(1) Correlation Coefficients

Techniques like Pearson [49] or Spearman correlation [50] coefficients are used to quantify the strength and direction of the relationship between each feature and the target. Features with high correlation to the target are typically retained, while those with low correlation are discarded.

(2) Mutual Information

This measure quantifies the amount of information that one variable contains about another. It is particularly useful for assessing nonlinear relationships and can help identify features that are highly informative for classification [51]. Table 1 summarizes various filter-based feature selection algorithms that utilize mutual information. It highlights whether each algorithm considers factors such as relevance, redundancy, complementarity, and interactions among features. Relevance refers to the statistical relationship between features, where highly correlated features may introduce redundancy. Redundancy occurs when multiple features provide overlapping information, which can negatively impact model efficiency. Complementarity describes how different features can collectively contribute unique information to improve predictive performance. Interactions among features capture complex dependencies, such as nonlinear relationships, that are crucial for understanding the underlying patterns in the data. By addressing these factors, feature selection algorithms aim to identify the most informative subset of features while minimizing noise and redundancy. The table also outlines the objective functions of these algorithms, providing researchers with a comparative framework to select appropriate feature selection methods.

Filter methods are computationally efficient and straightforward to implement. However, they have limitations, as they do not consider the interactions between features [61]. As a result, a feature that might be less significant in isolation could be critical in conjunction with other features, potentially leading to suboptimal selections.

#### 2.2.2. Wrapper Methods

Wrapper methods evaluate feature subsets based on the performance of a specific ML algorithm. These techniques involve a more iterative process, where various combinations of features are tested, and the performance of the model is assessed for each subset. Common wrapper methods include the following:

(1) Forward Selection

This approach starts with no features and adds them one by one based on their contribution to model performance. The process continues until adding more features no longer improves the model significantly [46,62].

(2) Backward Elimination

In contrast to forward selection, backward elimination begins with all available features and systematically removes the least significant ones until the model’s performance begins to degrade [63,64].

(3) RFE

Recursive feature elimination (RFE) employs a recursive approach to identify and remove the least important features based on model performance. It continues this process iteratively until the desired number of features is achieved [65,66].

(4) SVM-based

Support vector machines (SVMs) are often favored as an evaluation strategy due to their ability to deliver robust and high-accuracy results, especially in high-dimensional feature spaces. SVMs excel in finding the optimal decision boundary by maximizing the margin between classes, making them particularly effective for feature selection tasks where the goal is to identify the most relevant features for classification. Other popular algorithms such as decision trees (DTs), naïve Bayes (NB), K-nearest neighbors (KNN), artificial neural networks (ANNs), and linear discriminant analysis (LDA) are also commonly used in wrapper approaches. Table 2 displays the feature subset search strategies and evaluation criteria used by several wrapper-based algorithms. These methods utilize swarm intelligence algorithms as search strategies, such as particle swarm optimization (PSO) and genetic algorithms (GAs), which are particularly effective in exploring large, complex search spaces and are widely used in wrapper-based feature selection.

#### 2.2.3. Embedded Methods

Embedded methods integrate feature selection into the model training process, combining the advantages of both filter and wrapper methods. In these approaches, feature selection is performed as part of the model optimization. Embedded methods strike a balance between computational efficiency and classification performance, as they consider feature interactions while training the model. However, the choice of the underlying model can influence feature selection outcomes, necessitating careful consideration when applying these techniques.

(1) Regularization Techniques

Methods such as least absolute shrinkage and selection operator (LASSO, L1 regularization) [82,83] and ridge regression (L2 regularization) [84,85] penalize the complexity of the model by adding a penalty term. LASSO can shrink some feature coefficients to zero, effectively performing feature selection while fitting the model. Table 3 provides an overview of several classic LASSO-based feature selection algorithms, highlighting their key advantages and limitations, and offering insights into their applicability for different types of data and classification tasks.

(2) Decision Trees and Ensemble Methods

Algorithms like random forests (RFs) and gradient boosting inherently perform feature selection by evaluating the importance of features during the model-building process [25,86,87,88,89]. RF feature selection is an embedded method widely used in high-dimensional data analysis. Ranking features based on their contribution to model accuracy helps in identifying the most significant predictors. This method is particularly effective for data with many features [90,91]. Features that contribute more significantly to the model’s predictive power receive higher importance scores.

**Table 3 biomolecules-15-00081-t003:** Summary of classic LASSO-based feature selection algorithms with key advantages and limitations.

Name	Advantages	Limitations	Refs.
L1-regularized LASSO	(1) Helps avoid overfitting by shrinking coefficients. (2) Produces a sparse solution with fewer features.	(1) Can perform poorly if the relationship between features is highly nonlinear. (2) Might discard weak but informative features.	[92,93]
Elastic Net	(1) Combines LASSO and ridge regression, allowing for the selection of correlated features. (2) Performs well in cases of highly correlated data.	(1) Can be computationally more expensive than LASSO. (2) Requires careful tuning of the mixing parameter.	[94,95]
Sparse LASSO	(1) Provides a sparse solution, making it suitable for feature selection. (2) Useful when the number of predictors exceeds the number of observations.	(1) Can produce overly sparse models if the regularization is too strong. (2) May miss important features in some cases.	[96,97]
Group LASSO	(1) Allows for feature selection in groupings, useful when predictors can be naturally grouped (e.g., genes in bioinformatics). (2) Handles correlated variables within groups better than standard LASSO.	(1) Requires prior knowledge of the feature groupings. (2) Computationally more demanding.	[98,99]
Fused LASSO	(1) Good at identifying correlated groups of features that exhibit consistent changes. (2) Effective for time series or ordered data.	(1) Tends to perform poorly when there is no structure or correlation between features. (2) Limited flexibility for nonlinear relationships.	[100,101]
Adaptive LASSO	(1) More flexible and accurate than standard LASSO, as it adapts the penalty based on the data. (2) Can perform better when there is heterogeneity in feature selection.	(1) Requires the choice of an adaptive weight, which can be difficult to tune. (2) May overfit if not regularized properly.	[102,103]

#### 2.2.4. Swarm Intelligence

Swarm intelligence refers to the collective behavior of decentralized, self-organized systems, often observed in nature, such as the flocking behavior of birds, the schooling of fish, or the foraging behavior of ants [46,104]. These algorithms are widely used in feature selection due to their ability to effectively explore large search spaces and find optimal or near-optimal solutions [105]. The most used swarm intelligence algorithms in feature selection are PSO, GA, ant colony optimization (ACO), artificial bee colony (ABC), and simulated annealing (SA) algorithms. These swarm intelligence-based methods are particularly advantageous in wrapper-based feature selection approaches, where performance is evaluated by training a model on different feature subsets. Swarm intelligence algorithms help search the vast space of feature combinations more efficiently than exhaustive methods, making them valuable tools for high-dimensional data.

(1) PSO

PSO is inspired by the social behavior of birds flocking or fish schooling. Each particle (representing a potential solution) moves through the search space, adjusting its position based on both its own experience and the experiences of neighboring particles [73]. The algorithm is especially useful for optimizing complex, high-dimensional problems like feature selection, where traditional search methods might struggle.

(2) GA

GA is an evolutionary algorithm inspired by the process of natural selection and genetics [80]. In the context of feature selection, GA treats the feature subsets as “individuals” in a population, with everyone being a set of selected features. These individuals undergo processes like selection, crossover, and mutation to generate new feature subsets. The fitness of each subset is evaluated based on model performance (e.g., accuracy or error rate). Through generations, GA evolves better feature subsets that improve model performance.

(3) ACO

ACO is based on the foraging behavior of ants, where ants deposit pheromones that influence the decisions of other ants [106]. This technique has been applied to feature selection by simulating an ant colony that explores different feature subsets, guided by pheromone intensity. Over time, the colony converges on an optimal or near-optimal feature subset.

(4) ABC

ABC is a swarm intelligence optimization technique inspired by the foraging behavior of honeybees [106,107]. In this algorithm, artificial bees are divided into employed, onlooker, and scout bees, each performing different tasks to explore and exploit the search space. Employed bees search for food sources (solutions), onlooker bees choose food sources based on probability distribution, and scout bees randomly search for new solutions. The algorithm is particularly useful in solving optimization problems, such as feature selection, due to its ability to efficiently explore large solution spaces and converge towards optimal or near-optimal solutions.

(5) SA

SA is a probabilistic optimization algorithm inspired by the annealing process in metallurgy, where materials are heated and then slowly cooled to remove defects and reach a stable configuration [108,109]. SA mimics this process by starting with a random solution and iteratively exploring neighboring solutions. At each step, a new solution is accepted based on a temperature-dependent probability, allowing for occasional acceptance of worse solutions to escape local minima. Over time, the temperature decreases, and the algorithm converges to an optimal or near-optimal solution. SA is widely used for problems with large search spaces and complex objective functions, such as feature selection in ML.

### 2.3. Comparative Evaluation of Feature Selection Techniques: Advantages and Limitations

Selecting the most effective feature selection method requires a careful balance between efficiency, accuracy, and computational resources. Filter methods are valued for their swiftness and ability to handle high-dimensional data without relying on specific ML models. However, their primary shortcoming lies in ignoring the complex interplay between features, which could lead to suboptimal predictive accuracy. In contrast, wrapper methods, despite their high computational cost due to the need for repeated model training, provide a customized approach that accounts for feature interactions, yet they are prone to overfitting and are less effective with large datasets. Embedded methods strike a balance by integrating feature selection into the model training process, which boosts efficiency and performance, but their reliance on certain models and the complexity of their implementation can be restrictive. Swarm intelligence methods excel in their flexibility and ability to conduct global searches in high-dimensional spaces, but they are sensitive to parameter tuning and have higher computational expenses. Therefore, when aiming to optimize feature selection for cancer subtype classification, it is imperative to weigh the advantages and limitations of filter, wrapper, embedded, and swarm intelligence methods against the analytical objectives and the nature of the dataset in question. Table 4 contrasts these methods, highlighting their key benefits and constraints, to guide researchers in choosing the most suitable approach for their specific needs.

It is also worth noting that certain feature selection methods are particularly well-suited to specific machine learning algorithms. For example, LASSO is highly effective in regression problems due to its ability to shrink coefficients and perform feature selection through regularization, making it ideal for linear models. Similarly, RF excels in handling high-dimensional datasets by inherently ranking feature importance during model training. Methods like RFE work well with SVM because they iteratively refine feature subsets, optimizing for model performance. Furthermore, mutual information-based methods are particularly effective in capturing nonlinear relationships, which are beneficial for models like decision trees or kernel-based algorithms. These alignments highlight the importance of selecting feature selection methods that align with the characteristics of the machine learning algorithm used.

## 3. Applications of Feature Selection in Tumor Subtype Classification

### 3.1. Application Cases of Feature Selection Techniques in Tumor Subtype Classification

The application of feature selection techniques in tumor subtype classification is crucial for enhancing diagnostic accuracy, personalizing treatment strategies, and improving patient outcomes. Specifically, feature selection helps refine diagnostic models by identifying the most relevant biomarkers associated with specific tumor subtypes, reducing noise and enhancing model performance. This not only aids in accurate subtype classification but also enables the development of personalized prognostic models. These models support clinicians in tailoring treatment strategies to individual patients, ultimately leading to improved therapeutic efficacy and patient outcomes. By focusing on the most relevant features within high-dimensional datasets, researchers can better understand the molecular underpinnings of different tumor types and their associated behaviors [4,16]. The effectiveness of feature selection techniques has been underscored by various studies across different cancer types, as detailed in Table 5. For example, this approach has been successfully applied in the classification of breast cancer, a heterogeneous disease with distinct molecular subtypes such as Luminal A (LUMA), Luminal B (LUMB), HER2-Enriched, and basal-like (similar to the triple-negative breast cancer, TNBC), each with different biological characteristics and clinical implications. In addition, feature selection has been instrumental in lung cancer classification, the leading cause of cancer-related deaths globally, which is primarily divided into non-small cell lung cancer (NSCLC) and small cell lung cancer (SCLC). NSCLC, accounting for approximately 85% of all lung cancers, includes subtypes like lung adenocarcinoma (LUAD) and lung squamous cell carcinoma (LUSC). On the other hand, SCLC is known for its rapid growth and aggressive behavior, necessitating distinct therapeutic approaches. Moreover, renal cell carcinoma (RCC), the most common type of kidney cancer in adults, exhibits heterogeneity and can be subtyped into kidney renal clear cell carcinoma (KIRC), kidney renal papillary cell carcinoma (KIRP), and kidney chromophobe cell carcinoma (KICH) based on pathological and molecular features, each with differing genetic mutations, prognoses, and treatment strategies. As summarized in Table 5, the studies provide information on the cancer subtypes, the feature selection methods employed, the data types used, and the key biomarkers identified. These findings underscore the impact of feature selection on clinical outcomes, demonstrating its utility in improving the accuracy of tumor subtype classification and, by extension, the personalization of treatment and enhancement of patient care.

### 3.2. Impact of Tumor Subtype Classification on Clinical Outcomes

The impact of cancer subtype classification on clinical outcomes is profound, as it directly influences diagnostic precision, treatment decisions, and patient prognosis. By accurately identifying tumor subtypes, clinicians can tailor therapies to specific molecular profiles, improving treatment efficacy. For instance, certain subtypes of breast cancer, such as HER2-enriched tumors, are known to respond positively to targeted therapies like trastuzumab [135,136]. Conversely, subtypes like basal-like and TNBC require alternative treatment strategies due to their distinct molecular characteristics. This personalized treatment strategy has the potential to reduce adverse effects and enhance overall survival rates. Recent advancements in AI algorithms have further enhanced the precision of subtype classifications by integrating multi-dimensional datasets. These algorithms enable more accurate subtype identification, facilitating the selection of therapies that align with the molecular features of the tumor [137,138].

Moreover, robust subtype classification plays a pivotal role in predicting disease progression and guiding follow-up care. By distinguishing between aggressive and indolent subtypes, clinicians can make informed decisions about surveillance and treatment intensity. The incorporation of AI-driven models, particularly those utilizing ML and DL techniques, has proven effective in analyzing complex patterns within high-dimensional biomedical data, facilitating early detection and risk stratification [139,140]. In cases where early detection and subtype identification are pivotal—such as lung cancer—subtype classification can help identify high-risk patients who may benefit from more aggressive surveillance or early intervention, ultimately improving survival rates [141,142]. In oral diseases, subtype classification supported by ML techniques has provided novel insights into diagnosis and prognosis. These approaches have optimized treatment planning and reduced the potential for human error, ensuring that patients receive data-driven, personalized care [143]. Furthermore, AI applications have demonstrated promise in automating the identification of subtle molecular markers, supporting clinicians in making precise, evidence-based decisions that enhance patient outcomes.

## 4. Challenges and Limitations

Cancer subtype classification using feature selection techniques faces significant challenges, primarily stemming from the inherent complexity and heterogeneity of cancer. For instance, the high dimensionality and sparsity of omics data, where datasets often contain thousands of features (e.g., genes, proteins) but relatively few samples, increase the risk of overfitting. This imbalance can lead to models capturing noise rather than meaningful biological signals, reducing their generalizability [144]. Additionally, integrating multi-omics data, such as genomic, transcriptomic, and clinical information, poses difficulties due to variations in data quality, measurement platforms, and scales. Inappropriate data fusion may obscure meaningful relationships, compromising the accuracy and interpretability of classification models [4].

On the other hand, existing feature selection methodologies exhibit notable limitations that hinder their application. For example, while filter, wrapper, and embedded methods are widely used, each has its drawbacks—filter methods may overlook interaction effects between features, wrapper methods are computationally expensive and prone to overfitting in high-dimensional data, and traditional methods often struggle with feature correlation, resulting in unstable feature selection. This instability reduces confidence in the selected features, as multiple equally optimal signatures are frequently produced [20,145,146]. Moreover, the lack of standardized best practices reduces consistency and reproducibility across studies. Finally, a limitation of this review is its primary focus on structured/tabular data, which does not fully encompass the application of feature selection techniques to multimodal datasets. These datasets integrate structured (e.g., tabular), semi-structured (e.g., time series), and unstructured data (e.g., images, clinical notes) [147,148,149]. Future research should explore feature selection methods tailored to multimodal data, leveraging the complementary nature of diverse data types to uncover novel insights.

## 5. Future Directions

Future directions in cancer subtype classification using feature selection techniques can focus on several key areas. First, integrating multi-omics data holds significant promise. While most studies rely on a single data type (e.g., gene expression), combining genomic, transcriptomic, proteomic, and even clinical data can provide a more comprehensive understanding of tumor heterogeneity. This integrative approach could lead to more accurate and robust classification models, better capturing the complexity of cancer subtypes and their underlying biology. Second, addressing the challenge of imbalanced datasets is crucial for enhancing classification performance. Many cancer datasets suffer from small sample sizes and class imbalance, which can lead to overfitting and unreliable results. Advanced feature selection methods, such as cost-sensitive algorithms or hybrid models, could be developed to better handle these issues, ensuring that subtype classification remains accurate and applicable in clinical settings. Third, combining feature selection methods offers a promising avenue for future research. Hybrid strategies that integrate filter, wrapper, and embedded methods can capitalize on the strengths of each approach. For example, filter methods can efficiently reduce the dimensionality of large datasets, while wrapper methods can refine the selection process by evaluating subsets of features in relation to specific ML models. Such combinations may help improve the robustness, accuracy, and interpretability of predictive models. Moreover, there is a critical need to explore feature selection techniques capable of handling multimodal datasets. Integrating diverse data types could significantly enhance the interpretability and clinical utility of cancer subtype classification models. Finally, the incorporation of DL and AI-based models represents a frontier for improving cancer subtype classification. While traditional ML methods have proven effective, DL models, particularly those using convolutional neural networks (CNNs) or recurrent neural networks (RNNs), have the potential to automatically learn complex features from raw data. These models could offer improved performance in identifying subtle patterns and biomarkers, enabling more personalized cancer treatment strategies. By focusing on these areas, future research can further enhance the precision and applicability of cancer subtype classification, ultimately advancing personalized medicine.

## 6. Conclusions

In this review, we explored various feature selection methodologies, including filter, wrapper, and embedded techniques, each with its strengths and limitations. Case studies across diverse cancer types illustrate the practical application of these techniques, showcasing their potential to uncover critical biomarkers that can inform clinical decision-making. The integration of AI into biomedicine has been a cornerstone, underscoring its transformative role in enhancing feature selection and tumor subtype classification across various biomedical applications.

However, the journey toward optimized feature selection is fraught with challenges, including issues related to data quality, interindividual variability, and the risk of overfitting. Addressing these challenges is crucial for ensuring that feature selection outcomes are reliable, reproducible, and clinically relevant, which is of paramount importance in the field of AI in biomedicine. Looking ahead, the integration of multi-omics data and advancements in ML offer promising avenues for enhancing feature selection and tumor subtype classification. These innovations can lead to more comprehensive biomarker discovery and improved model performance, which are key areas of focus in AI applications in bioinformatics and molecular biology. As we continue to unravel the complexities of tumor biology, the role of feature selection will remain pivotal in advancing personalized medicine, improving patient outcomes, and enhancing the overall landscape of cancer care.

## Figures and Tables

**Figure 1 biomolecules-15-00081-f001:**
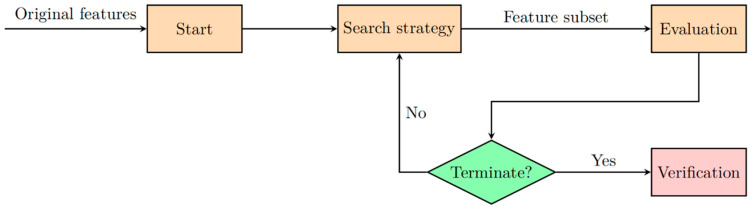
The workflow diagram of feature selection techniques.

**Figure 2 biomolecules-15-00081-f002:**
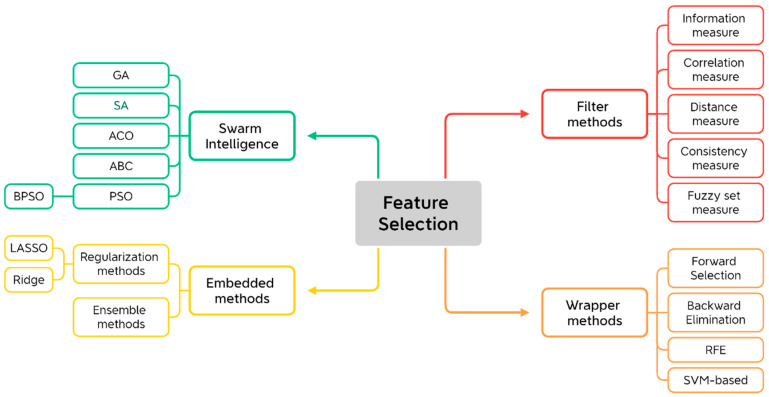
Overall categories of feature selection techniques. RFE: recursive feature elimination; SVM: support vector machines; LASSO: least absolute shrinkage and selection operator; GA: genetic algorithms; SA: simulated annealing; ACO: ant colony optimization; ABC: artificial bee colony; PSO: particle swarm optimization; BPSO: binary particle swarm optimization.

**Table 1 biomolecules-15-00081-t001:** Overview of filter-based feature selection algorithms based on mutual information.

Algorithm	Relevance	Redundancy	Complementarity	Interactivity	Objective Function
MIM [52]	✓	✗	✗	✗	$\arg\max_{f_{k}\in F-S}I(f_{k};C)$
MIFS [53]	✓	✓	✗	✗	$\arg\max_{f_{k}\in F-S}(I(f_{k};C)-\beta \sum_{f_{j}\in S}I(f_{k};f_{j}))$
mRMR [54]	✓	✓	✗	✗	$\arg\max_{f_{k}\in F-S}(I(f_{k};C)-\frac{1}{\left|S\right|}\sum_{f_{j}\in S}I(f_{k};f_{j}))$
CMIM [55]	✓	✓	✗	✗	$\arg\max_{f_{k}\in F-S}(\min_{f_{j}\in S}I(f_{k};C|f_{j}))$
JMI [56]	✓	✓	✗	✗	$\begin{array}{c} \arg\max_{f_{k}\in F-S}(I(f_{k};C)-\frac{1}{\left|S\right|}\sum_{f_{j}\in S}I(f_{k};f_{j})\\ +\frac{1}{\left|S\right|}\sum_{f_{j}\in S}I(f_{k};f_{j}|C)) \end{array}$
CIFE [57]	✓	✓	✗	✗	$\begin{array}{l} \arg\max_{f_{k}\in F-S}(I(f_{k};C)-\sum_{f_{j}\in S}I(f_{k};f_{j})\\ +\sum_{f_{j}\in S}I(f_{k};f_{j}|C)) \end{array}$
MRI [58]	✓	✓	✓	✗	$\begin{array}{l} \arg\max_{f_{k}\in F-S}(I(f_{k};C)\\ +\sum_{f_{j}\in S}\{I(f_{j};C|f_{k})+I(f_{k};C|f_{j})\}) \end{array}$
MRMI [59]	✓	✓	✗	✓	$\begin{array}{l} \arg\max_{f_{k}\in F-S}(I(f_{k};C)\\ +\max_{f_{i}\in F-S}(\frac{I(f_{k};f_{j};C)}{I(f_{k};f_{j})}-\frac{I(f_{k};f_{j};C)}{I(f_{j};C)})) \end{array}$
DISR [60]	✓	✓	✓	✓	$\arg\max_{f_{k}\in F-S}(\sum_{f_{j}\in S}I(f_{k},f_{j};C)/H(f_{i},f_{s};C))$

MIM: mutual information maximization; MIFS: mutual information for selecting features; mRMR: minimal redundancy maximal relevance; CMIM: conditional mutual information maximization; JMI: joint mutual information; CIFE: conditional informative feature extraction; MRI: max-relevance and max-independence; MRMI: minimum redundancy and maximum interaction; DISR: double input symmetrical relevance.

**Table 2 biomolecules-15-00081-t002:** Search strategies and evaluation criteria of some wrapper-based methods.

Algorithms	Search Strategy	Evaluation Criteria	Ref.
LSEFS	PSO	SVM	[67]
Hybrid feature selection	PSO, GA	SVM	[68]
GA-based feature selection	GA	SVM	[69]
mr^2^PSO	PSO	SVM	[70]
GA-based feature selection	GA	SVM	[71]
PSO–SVM	PSO	SVM	[72]
PSO–SVM	PSO	SVM	[73]
EAwPS	GA	KNN	[74]
GA-based feature selection	GA	KNN	[75]
PSO-based feature selection	BPSO	KNN	[76]
IniPG	PSO	KNN	[77]
GA-based feature selection	GA	NB	[78]
GA-based feature selection	GA	NB	[79]
GA-based feature selection	GA	NB	[80]
GRASP	GA	ANN	[81]

LSEFS: LSSVM-based evolutionary feature selection; PSO: particle swarm optimization; SVM: support vector machines; GA: genetic algorithm; mr2PSO: maximum relevance minimum redundancy PSO; EAwPS: evolutionary algorithms with a partial sequential forward floating search mutation; KNN: K-nearest neighbors; IniPG: initialization strategy and updating mechanism based on PSO and genetic programming; NB: naïve Bayes; GRASP: greedy randomized adaptive search procedures; ANN: artificial neural network.

**Table 4 biomolecules-15-00081-t004:** Key advantages and limitations of four types of feature selection methods.

Algorithm	Advantages	Limitations	Refs.
Filter methods	(1) Fast and computationally efficient. (2) Scalable for high-dimensional datasets. (3) No dependency on ML models. (4) Suitable for preprocessing large datasets.	(1) Ignores feature interactions and dependencies. (2) May select suboptimal features for predictive accuracy. (3) Not ideal for complex relationships between features.	[20,110]
Wrapper methods	(1) Directly evaluates model performance, ensuring better feature subset selection. (2) Considers feature interactions and dependencies. (3) Can be tailored to specific models or tasks.	(1) Computationally expensive due to repeated model training. (2) Prone to overfitting, especially with small datasets. (3) Poor scalability with large datasets and many features.	[111,112,113]
Embedded methods	(1) Feature selection is integrated during model training, making it more efficient. (2) Optimizes features to improve model performance directly. (3) Less computational overhead compared to wrapper methods.	(1) Typically model-dependent, limiting flexibility across different algorithms. (2) More complex to implement. (3) Limited to models that support feature selection (LASSO, decision trees).	[114,115]
Swarm intelligence	(1) Effective in complex, nonlinear, and high-dimensional problems. (2) Global search capability avoids local optima. (3) Highly flexible and adaptable to different problems.	(1) Computationally expensive due to the iterative search process. (2) Sensitive to parameter settings (e.g., particle size, mutation rates). (3) Potential convergence issues and lack of theoretical guarantees.	[46,71,105]

**Table 5 biomolecules-15-00081-t005:** Summary of studies utilizing feature selection techniques for cancer subtype classification.

Tumor Organ	Tumor Subtype	Data Type	Methodology	Biomarkers or Features	Refs.
Lung	LUAD, LUSC	Gene expression	Gradient boosting	SFTA2, TRIM29, AKR1B0, KRT5, PKP1	[116]
Lung	LUAD, LUSC	Gene expression	RF	KRT17, KRT14, KRT6A, TRIM29, KRT5, NECTIN1, TUBA1C, S100A2	[117]
Lung	LUAD, LUSC	Proteomic data	Weight-based feature selection	TFRC, BRD4, CD26, INPP4B, IGFBP2, DUSP4	[118]
Glioma	LGG, HGG	Proteomic data	pyHSICLasso, XGBoost, RF	CYCLIND1, CYCLINE2, ERK2, IGF1R_pY1135Y1136, PAI1, PDK1, PR	[119]
Kidney	KIRC, KIRP, KICH	Proteomic data	pyHSICLasso, XGBoost, RF	CKIT, FASN, MEK1, PR, ARAF_pS299	[119]
Lung	LUAD, LUSC	Proteomic data	pyHSICLasso, XGBoost, RF	INPP48, DUSP4, MIG6, CD26, TFRC, NF2, HER3	[119]
Lung	LUAD, LUSC	Radiomic data	LASSO	LGSRE, HGZE, ZP, IDMCM, LNE	[120]
Breast	TNBC, Others	Imaging data	RF, SHAP	Mass_Indistinct, Mass_Spiculated, US_Mass_one_para, Calc_amorphous	[121]
Breast	Basal, LUMB	Proteomic data	Wrapper method, spectral clustering	CENPU, KIAA0101, NUSAP1, PBK, RRM2, TOP2A	[122]
Breast	TNBC, Others	Circulating miRNAs	Ensemble recursive feature selection	hsa-miR-378, hsa-miR-221, hsa-miR-630, hsa-miR-145, hsa-miR-342-3p	[39]
Breast	Basal, HER2-enriched, LUMA, LUMB, Normal-like	Omics data	Relevance–Redundancy assessment (ReRa)	-	[123]
Lung	LUAD, LUSC	Radiomic data	mRMR, SFS, LASSO	wavelet-LLH _ firstorder _ Skewness, Wavelet-HHL _ glcm _ ClusterShade	[124]
Breast	LUMA, LUMB, HER2-enriched, Basal	miRNA expression	Ensemble of 8 feature selection methods (e.g., MIM, mRMR)	hsa-miR-25-3p, hsa-miR-505-5p, hsa-miR-29b-2-5p, hsa-miR-10a-5p, hsa-miR-140-3p, hsa-miR-30c-2-3p, hsa-miR-193a-5p	[125]
Breast	LUMA, LUMB, Basal, ERBB2	miRNA expression	11 meta-heuristic algorithms (e.g., PSO, GA)	miR-135, miR-188, miR-449, miR-29, miR-101, miR-105, miR-190, miR-33	[126]
Breast	TNBC, Non-TNBC	Gene expression	False discovery rates (FDRs) gene selection	ESR1, MLPH, FSIP1, C5AR2, GATA3, TBC1D9, CT62, TFF1, PRR15, CA12, AGR3	[127]
Breast	LUMA, LUMB, Basal, HER2-enriched	Gene expression	Forest subtype	PCAT29, GATA3, CCDC170, SPDEF, SLC7A13, BIRC5, SPAG5, C5AR2	[128]
Multi-cancer	Multi-subtypes	Single nucleotide variants	Multi-dimensional SNVs feature definition	-	[129]
Breast	TNBC, Non-TNBC	Clinicopathological data	GA, SVM-RFE, LASSO	-	[130]
Kidney	KIRC, KIRP, KICH	mRNA expression, lncRNA expression	Sequential reinforcement active feature learning (SRAFL)	LINC00887, TTC21B-AS1, SLC47A1P1, SLC10A2, AL109946.1, UQCRB, OR2T10, ENPP7P8	[131]
Leukemia	ALL, AML	Gene expression	Transductive SVM (TSVM)	M27891_at, Y07604_at	[132]
SRBCT	EWS, NB, BL, RMS	Gene expression	TSVM	784224, 812105, 207274, 782811, 344134	[132]
MLL	ALL, MLL, AML	Gene expression	TSVM	31375_at, 31385_at, 31394_at, 31441_at	[132]
DLBCL	DLBCL, FL	Gene expression	TSVM	M59829_at, X53961_at, U46006_s_at, X85785_rna1_at	[132]
Leukemia	ALL, AML	Gene expression	Self-organizing maps (SOMs), fuzzy C-means clustering (FCC), Fisher’s linear discriminant	-	[133]
Brain	MD, Mglio, Rhab, PNET, Ncer	Gene expression	SOM, FCC, Fisher’s linear discriminant	-	[133]
Lung	LADC, SQCLC, SCLC	DNA methylation	mRMR, RF	-	[32]
Breast	LUMA, LUMB, Basal, HER2-enriched	Gene expression	Outlier-based gene selection (OGS)	AGR2, AGR3, EN1, FOXA1, FOXC1, FZD9, KIAA1324, PRR15, SPDEF, TMC5, C1orf106, CEACAM5, FBXO10, GRIK3, GRPR	[134]

For tumor organ, SRBCT: small round blue-cell tumor; MLL: mixed-lineage leukemia; DLBCL: diffuse large B-cell lymphoma. For tumor subtype, LUAD: lung adenocarcinoma; LUSC: lung squamous cell carcinoma; LGG: low-grade glioma; HGG: high-grade glioma; KIRC: kidney renal clear cell carcinoma; KIRP: kidney renal papillary cell carcinoma; KICH: kidney chromophobe cell carcinoma; TNBC: triple-negative breast cancer; LUMA: Luminal A; LUMB: Luminal B; ALL: acute lymphoblastic leukemia; AML: acute myeloid leukemia; EWS: Ewing’s sarcoma; NB: neuroblastoma; BL: Burkitt’s lymphoma; RMS: rhabdomyosarcoma; DLBCL: diffuse large B-cell lymphomas; FL: follicular lymphoma; MD: medulloblastoma; Mglio: malignant glioma; Rhab: atypical teratoid/rhabdoid tumor; PNET: primitive neuroectodermal tumor; Ncer: normal cerebella tumor; LADC: lung adenocarcinoma; SQCLC: squamous cell lung cancer; SCLC: small cell lung cancer.

## Data Availability

Not applicable.

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
