# Peer review of "Utilizing Feature Selection Techniques for AI-Driven Tumor Subtype Classification: Enhancing Precision in Cancer Diagnostics"

_biomolecules, 2025, doi:10.3390/biom15010081_

Round 1
Reviewer 1 Report
Comments and Suggestions for Authors
The authors present a well-written and comprehensive overview of feature selection. There are a few points that should be clarified.
Please add a brief definition of feature selection, deep learning, and machine learning for the general audience where the terms are first used in the Introduction.
Did the authors include “Random Forest Feature Selection”?
Some feature selection methods work well with specific machine learning methods. Can the authors comment on this?
Some machine learning methods seem to work better with certain types of data. For example RF works well with data with lost of features. Is this the case with the feature selection methods?
Can the authors comment of combining feature selection methods?
Deep learning and machine learning are quite different. While deep learning is good for image data, is deep learning really relevant in the exploration of multiomic data. Does DL use the same feature selection at ML?
Author Response
Comments and Suggestions for Authors
The authors present a well-written and comprehensive overview of feature selection. There are a few points that should be clarified.
Answer: We thank the Reviewer very much for the positive comments and valuable suggestions for our Manuscript, we have modified the content according to the Reviewers’ suggestions, with the revisions are marked in red. We hope that the quality of the revised manuscript will be improved.
- Please add a brief definition of feature selection, deep learning, and machine learning for the general audience where the terms are first used in the Introduction.
Answer: We thank the Reviewer for the careful review, and the requested definitions have been added in the revised Manuscript to ensure clarity for a general audience:
Feature selection refers to the process of identifying the most informative variables from high-dimensional datasets to reduce dimensionality while retaining essential information.
ML is a branch of AI that involves training algorithms to recognize patterns and make predictions based on data. DL, a subset of ML, uses multi-layered neural networks to automatically learn hierarchical feature representations from raw data.
- Did the authors include “Random Forest Feature Selection”?
Answer: We thank the Reviewer for the valuable suggestion. In the revised Manuscript, we added a detailed description of Random Forest (RF) Feature Selection under embedded methods:
RF feature selection is an embedded method widely used in high-dimensional data analysis. By ranking features based on their contribution to model accuracy, it helps in identifying the most significant predictors. This method is particularly effective for data with many features.
- Some feature selection methods work well with specific machine learning methods. Can the authors comment on this?
Answer: We appreciate the valuable comments very much. In the revised Manuscript, we added an analysis of how specific feature selection methods align with certain machine learning models in the subsection of 2.3. Comparative Evaluation of Feature Selection Techniques: Advantages and Limitations, to enhance predictive accuracy:
It is also worth noting that certain feature selection methods are particularly well-suited to specific machine learning algorithms. For example, LASSO is highly effective in regression problems due to its ability to shrink coefficients and perform feature selection through regularization, making it ideal for linear models. Similarly, RF excels in handling high-dimensional datasets by inherently ranking feature importance during model training. Methods like RFE work well with SVM because they iteratively refine feature subsets, optimizing for model performance. Furthermore, mutual information-based methods are particularly effective in capturing nonlinear relationships, which are beneficial for models like decision trees or kernel-based algorithms. These alignments highlight the importance of selecting feature selection methods that align with the characteristics of the machine learning algorithm used.
- Some machine learning methods seem to work better with certain types of data. For example, RF works well with data with lost of features. Is this the case with the feature selection methods?
Answer: We thank the Reviewer for the insightful comment. Indeed, just as certain machine learning methods are better suited to specific types of data, feature selection methods also exhibit varying levels of effectiveness depending on the characteristics of the data. For instance, mutual information-based methods are particularly effective for datasets with nonlinear relationships, as they capture complex dependencies between features and the target variable. On the other hand, filter methods such as correlation-based techniques are advantageous for preprocessing large datasets due to their computational efficiency, especially when dealing with high-dimensional data.
Additionally, methods like RF feature selection excel in scenarios with many features, as the algorithm inherently evaluates feature importance during training. These observations underscore the importance of aligning feature selection techniques with the structure and nature of the data to achieve optimal performance.
- Can the authors comment of combining feature selection methods?
Answer: We agree with the Reviewer very much and have added comments of combining feature selection methods in the context of 5. Future Directions:
Third, combining feature selection methods offers a promising avenue for future research. Hybrid strategies that integrate filter, wrapper, and embedded methods can capitalize on the strengths of each approach. For example, filter methods can efficiently reduce the dimensionality of large datasets, while wrapper methods can refine the selection process by evaluating subsets of features in relation to specific ML models. Such combinations may help improve the robustness, accuracy, and interpretability of predictive models.
- Deep learning and machine learning are quite different. While deep learning is good for image data, is deep learning really relevant in the exploration of multiomic data. Does DL use the same feature selection at ML?
Answer: We thank the Reviewer for the constructive question. Deep learning (DL) and machine learning (ML) indeed differ significantly in their approaches, particularly regarding feature selection. While ML typically relies on explicit feature selection methods to identify the most informative features before model training, DL often bypasses this step. Instead, DL models, such as convolutional neural networks (CNNs) or recurrent neural networks (RNNs), automatically learn hierarchical feature representations directly from raw data during the training process.
In the context of multi-omics data, DL has demonstrated relevance and potential. Multi-omics datasets are inherently complex and high-dimensional, making them suitable for DL's ability to capture intricate patterns and nonlinear relationships. However, the applicability of DL in multi-omics research is still emerging and may require careful adaptation to address challenges such as limited sample sizes and data integration.

Reviewer 2 Report
Comments and Suggestions for Authors
The study provides a comprehensive non-systematic review of the role of feature selection techniques in AI-based tumor subtype classification.
Consider the following points:
1. Abstract is poor.
a) Consider "Cancer's heterogeneity poses challenges in accurate diagnosis and treatment, necessitating precise tumor subtype classification. This review explores the pivotal role of feature selection techniques in addressing these challenges,..."; what challenges?
b) The research topic should be introduced, including feature selection and its role on "Enhancing Precision in Cancer Diagnostics". There is no direct link, and this should be explained.
c) Consider "...while also addressing limitations like data quality, overfitting, and scalability"; Limitations of what and for what?
2. Consider "In this review, we systematically explored the role of feature selection techniques in unraveling tumor complexity and improving subtype classification." How did you systematically explore it? What do you mean by "systematically"? This is not a systematic review.
3. What is the novelty of this review when compared with others? For example:
https://www.sciencedirect.com/science/article/pii/S0010482519302525
https://www.frontiersin.org/journals/bioinformatics/articles/10.3389/fbinf.2022.927312/full
https://ieeexplore.ieee.org/document/9452069
The authors are recommended to present related work and highlight the novelty of the review.
4. The manuscript would benefit from examples for some explanations. For example, consider "This not only enhances the performance of classification algorithms but also aids in interpreting the underlying biological significance of selected features"; how does it enhance the performance and aid in the model interpretation? An example could enrich the explanation.
5. In section 2.2., you categorize methods into 4 approaches, but only in 3 previously (lines 52-63).
6. Avoid short paragraphs (e.g., lines 143-146).
7. Consider "It highlights whether each algorithm considers factors such as correlation, redundancy, complementarity, and interactions among features."; explain each factor.
8. Since you are explaining concepts, avoid comments to argue why the explanation is important. For example, "This information enhances the understanding of the characteristics and advantages of different algorithms, facilitating their effective application in data analysis and ML."
9. Why didn't you present all the techniques described in Figure 1 throughout the section? Only a few were presented.
10. What does the column 'Refs' mean in tables 3 and 4?
11. Consider "The application of feature selection techniques in tumor subtype classification is crucial for enhancing diagnostic accuracy, personalizing treatment strategies, and improving patient outcomes." What do you mean in this sentence? Are you talking about prognostic or diagnostic models? How do 'feature selection techniques in tumor subtype classification' improve patient outcomes?
12. As this is not a systematic review, how did you select the 'studies reviewed'?
13. Why are some columns in Table 5 empty?
14. Section 3.2 is poor. It does not present the 'Impact of Tumor Subtype Classification on Clinical Outcomes', neither qualitatively nor quantitatively. It just presents a general discussion on the topic. There is no analysis of the 'studies reviewed'.
15. In section 4. Challenges and Limitations, which are challenges and which are limitations? Is there any difference?
16. Are there only 3 challenges/limitations in this research area?
17. Figure 3 is not well described in section 5.
18. Do the authors consider the scope of the review to be limited to structured/tabular data? I think yes. So, my recommendation is to make this clear in the title and study objective.
19. I think the authors did not consider multimodality (multimodal datasets with structured, semi-structured, and unstructured data, such as time series, images texts, etc) in the review. However, there is a growing number of studies in this area, hence feature selection techniques should consider multimodal approaches. What do the authors think? See the following papers:
https://www.frontiersin.org/journals/artificial-intelligence/articles/10.3389/frai.2024.1408843/full
https://www.sciencedirect.com/science/article/pii/S2772941923000212
https://www.nature.com/articles/s43018-022-00388-9
Author Response
Comments and Suggestions for Authors
The study provides a comprehensive non-systematic review of the role of feature selection techniques in AI-based tumor subtype classification.
Answer: We thank the Reviewer very much for the positive comments and valuable suggestions for our Manuscript, we have modified the content according to the Reviewers’ suggestions, with the revisions are marked in red. We believe that the quality of the revised manuscript will be improved.
Consider the following points:
- Abstract is poor.
- a) Consider "Cancer's heterogeneity poses challenges in accurate diagnosis and treatment, necessitating precise tumor subtype classification. This review explores the pivotal role of feature selection techniques in addressing these challenges,..."; what challenges?
- b) The research topic should be introduced, including feature selection and its role on "Enhancing Precision in Cancer Diagnostics". There is no direct link, and this should be explained.
- c) Consider "...while also addressing limitations like data quality, overfitting, and scalability"; Limitations of what and for what?
Answer: We thank the Reviewer very much for the detailed comments and suggestions. We have revised the abstract to address the concerns:
Cancer's heterogeneity presents significant challenges in accurate diagnosis and effective treatment, including the complexity of identifying tumor subtypes and their diverse biological behaviors. This review examines how feature selection techniques address these challenges by improving the interpretability and performance of machine learning (ML) models in high-dimensional datasets. Feature selection methods—such as filter, wrapper, and embedded techniques—play a critical role in enhancing the precision of cancer diagnostics by identifying relevant biomarkers. The integration of multi-omics data and ML algorithms facilitates a more comprehensive understanding of tumor heterogeneity, advancing both diagnostics and personalized therapies. However, challenges such as ensuring data quality, mitigating overfitting, and addressing scalability remain critical limitations of these methods. Artificial intelligence (AI)-powered feature selection offers promising solutions to these issues by automating and refining the feature extraction process. This review highlights the transformative potential of these approaches while emphasizing future directions, including the incorporation of deep learning (DL) models and integrative multi-omics strategies for more robust and reproducible findings.
- Consider "In this review, we systematically explored the role of feature selection techniques in unraveling tumor complexity and improving subtype classification." How did you systematically explore it? What do you mean by "systematically"? This is not a systematic review.
Answer: We thank the reviewer for pointing this out. We acknowledge that the use of the term "systematically" may create confusion, as this is not a systematic review in the strict sense of the term. Our intention was to convey that we comprehensively analyzed and discussed various feature selection techniques, their applications, and their implications in cancer subtype classification, drawing on relevant literature and examples. To address this, we revised the sentence to avoid potential misunderstanding:
In this review, we provide a comprehensive exploration of feature selection techniques, their role in unraveling tumor complexity, and their contributions to improving subtype classification.
- What is the novelty of this review when compared with others? For example:
https://www.sciencedirect.com/science/article/pii/S0010482519302525
https://www.frontiersin.org/journals/bioinformatics/articles/10.3389/fbinf.2022.927312/full
https://ieeexplore.ieee.org/document/9452069
The authors are recommended to present related work and highlight the novelty of the review.
Answer: We thank the Reviewer for the valuable comments and suggestions. Indeed, these works highlight important aspects of feature selection: one focuses on its application in medical settings, emphasizing dimensionality reduction, disease understanding, and real-world case studies; another discusses how feature selection addresses the curse of dimensionality in machine learning models for disease risk prediction using genetic data; and a third explores feature selection, dimensionality reduction, and classification techniques for chronic disease diagnosis, emphasizing their impact on predictive accuracy and computational efficiency. Building upon these studies, our review uniquely focuses on the role of feature selection techniques in cancer subtype classification. We emphasize the integration of feature selection with AI-driven approaches and multi-omics data, offering insights into how these methods address challenges such as tumor heterogeneity and high-dimensional datasets. Furthermore, we discuss emerging trends, such as hybrid methods and deep learning models, providing a forward-looking perspective that distinguishes our review from existing literature.
We have included some introduction of these related works in the revised Manuscript to contextualize our contributions and highlight the novelty of this review:
Previous reviews have addressed feature selection in various contexts. For example, one review highlights its role in dimensionality reduction and disease understanding through real-world medical case studies [40]. Another explores how feature selection mitigates the curse of dimensionality in disease risk prediction using genetic data [41]. A third survey examines feature selection alongside dimensionality reduction and classification techniques for chronic disease diagnosis, emphasizing their impact on predictive accuracy and computational efficiency [42]. Building upon these studies, our review focuses specifically on the role of feature selection techniques in cancer subtype classification. In this review, we provide a comprehensive exploration of feature selection techniques, their role in unraveling tumor complexity, and their contributions to improving subtype classification.
- The manuscript would benefit from examples for some explanations. For example, consider "This not only enhances the performance of classification algorithms but also aids in interpreting the underlying biological significance of selected features"; how does it enhance the performance and aid in the model interpretation? An example could enrich the explanation.
Answer: We thank the Reviewer for the constructive suggestions. We added an example in the revised Manuscript:
For instance, study presents a machine learning framework that integrates somatic mutation and gene expression data to classify breast cancer subtypes. The authors employ feature selection techniques to identify key genes associated with specific subtypes. These selected features improved model accuracy by reducing noise and overfitting while also providing biologically meaningful insights into the molecular mechanisms of breast cancer [43].
- In section 2.2., you categorize methods into 4 approaches, but only in 3 previously (lines 52-63).
Answer: We thank the Reviewer for the valuable comment and careful review. In the Introduction, we focused on the three most conventional and fundamental types of feature selection methods—filter, wrapper, and embedded methods—to provide readers with a clear and foundational understanding of the topic. These three categories are widely recognized as the cornerstone of feature selection techniques and are often used as a starting point for discussion.
In the main body of the manuscript, we extended this classification by including swarm intelligence methods. This addition reflects the growing popularity and relevance of these methods in recent years, particularly in the context of integrating artificial intelligence (AI). Swarm intelligence methods often represent hybrid approaches that combine the strengths of the three conventional techniques, making them a distinct and innovative category worthy of separate discussion. Highlighting this fourth category allows us to better capture the evolving landscape of feature selection techniques and their applications in cancer research.
We have clarified this progression from conventional to advanced methods in the revised Manuscript, to improve the coherence between the Introduction and the main text:
In this review, we category feature selection into four primary approaches: filter methods [23,27], wrapper methods [37,44], embedded methods [25,45] and swarm intelligence methods [46,47], and each of these approaches has its unique advantages, limitations, and applications in cancer research, as shown in Figure 2. Here, we introduce swarm intelligence methods to reflect their increasing importance in modern applications [48]. Swarm intelligence algorithms are often viewed as an organic combination of the three conventional feature selection methods—filter, wrapper, and embedded techniques. These algorithms leverage the strengths of each approach by combining the statistical evaluation of features (from filter methods), the model performance optimization (from wrapper methods), and the integration of feature selection within the model training process (from embedded methods).
- Avoid short paragraphs (e.g., lines 143-146).
Answer: We thank the Reviewer for the comments. The short paragraph was intended to emphasize a specific point within the context of the manuscript. We aim to maintain clarity while ensuring the structure aligns with academic writing conventions.
Thank you again for your understanding.
- Consider "It highlights whether each algorithm considers factors such as correlation, redundancy, complementarity, and interactions among features."; explain each factor.
Answer: We thank the Reviewer very much for the careful review, and have added the explanation for each factor in the revised Manuscript:
Table 1 summarizes various filter-based feature selection algorithms that utilize mutual information. It highlights whether each algorithm considers factors such as relevance, redundancy, complementarity, and interactions among features. Relevance refers to the statistical relationship between features, where highly correlated features may introduce redundancy. Redundancy occurs when multiple features provide overlapping information, which can negatively impact model efficiency. Complementarity describes how different features can collectively contribute unique information to improve predictive performance. Interactions among features capture complex dependencies, such as nonlinear relationships, that are crucial for understanding the underlying patterns in the data. By addressing these factors, feature selection algorithms aim to identify the most informative subset of features while minimizing noise and redundancy.
- Since you are explaining concepts, avoid comments to argue why the explanation is important. For example, "This information enhances the understanding of the characteristics and advantages of different algorithms, facilitating their effective application in data analysis and ML."
Answer: We thank the Reviewer and agree with the Reviewer, we deleted the comments of "This information enhances the understanding of the characteristics and advantages of different algorithms, facilitating their effective application in data analysis and ML." in the revised Manuscript.
- Why didn't you present all the techniques described in Figure 1 throughout the section? Only a few were presented.
Answer: We thank the Reviewer for the careful review and we suppose what the Reviewer should say is Figure 2. Feature selection is indeed a vast topic encompassing a wide range of methods, as depicted in Figure 2. However, due to space limitations, it was not feasible to provide an exhaustive discussion of all these techniques within the scope of this manuscript. Instead, we focused on a subset of methods that are most relevant to the primary objective of the study, which is to explore the application of feature selection in cancer classification.
Our aim was not to provide a comprehensive review of feature selection methods but rather to highlight those approaches that are particularly impactful and widely used in the context of cancer research. By narrowing the focus, we were able to provide a more in-depth discussion of these methods and their integration into cancer classification tasks, which aligns with the central theme of the manuscript.
- What does the column 'Refs' mean in tables 3 and 4?
Answer: The 'Refs' mean 'References’.
- Consider "The application of feature selection techniques in tumor subtype classification is crucial for enhancing diagnostic accuracy, personalizing treatment strategies, and improving patient outcomes." What do you mean in this sentence? Are you talking about prognostic or diagnostic models? How do 'feature selection techniques in tumor subtype classification' improve patient outcomes?
Answer: We thank the Reviewer for the insightful comment. To clarify, the sentence refers to both diagnostic and prognostic models. Feature selection techniques play a critical role in tumor subtype classification by identifying the most relevant biomarkers associated with specific subtypes. This enhances diagnostic accuracy by refining models to reduce noise and improve predictive performance. Additionally, these techniques contribute to the development of personalized prognostic models, which support clinicians in tailoring treatment strategies to individual patients. By aligning treatments with the unique characteristics of each tumor subtype, these models can improve therapeutic efficacy and, consequently, patient outcomes.
We have revised the manuscript to clarify these points and provide a more comprehensive explanation:
Specifically, feature selection helps refine diagnostic models by identifying the most relevant biomarkers associated with specific tumor subtypes, reducing noise and enhancing model performance. This not only aids in accurate subtype classification but also enables the development of personalized prognostic models. These models support clinicians in tailoring treatment strategies to individual patients, ultimately leading to improved therapeutic efficacy and patient outcomes.
- As this is not a systematic review, how did you select the 'studies reviewed'?
Answer: We thank the Reviewer for the strict review of the manuscript. We have modified the sentence in the revised Manuscript:
As summarized in Table 5, the studies provide information on the cancer subtypes, the feature selection methods employed, the data types used, and the key biomarkers identified.
- Why are some columns in Table 5 empty?
Answer: We thank the Reviewer for the careful observation in Table 5. The empty cells indicate instances where specific information was not explicitly reported or available in the original sources. We included these columns to maintain consistency in the table structure and to highlight areas where data may be incomplete.
- Section 3.2 is poor. It does not present the 'Impact of Tumor Subtype Classification on Clinical Outcomes', neither qualitatively nor quantitatively. It just presents a general discussion on the topic. There is no analysis of the 'studies reviewed'.
Answer: We thank the Reviewer for the valuable feedback. We have revised the section on the impact of tumor subtype classification on clinical outcomes. The revised text elaborates on how precise subtype classification informs treatment decisions, predicts disease progression, and enhances patient care through the integration of AI-driven models and personalized approaches.
- In section 4. Challenges and Limitations, which are challenges and which are limitations? Is there any difference?
Answer: We thank the Reviewer for the careful review. In Section 4 of the manuscript, we have used the terms interchangeably to discuss both the inherent difficulties (challenges) in cancer subtype classification and the shortcomings (limitations) of existing feature selection techniques. For example:
Challenges: These refer to broader, systemic issues, such as the complexity and heterogeneity of cancer, the high dimensionality and sparsity of omics data, and the integration of multi-omics datasets. These represent unresolved problems that require innovative solutions.
Limitations: These are more specific to the methodologies used, such as the computational inefficiencies of wrapper methods, the instability of traditional feature selection methods, and the absence of standardized best practices.
- Are there only 3 challenges/limitations in this research area?
Answer: Thank you for raising this question. While the original manuscript discusses three key challenges and limitations—high dimensionality and sparsity of omics data, selection of appropriate feature selection methods, and integration of multi-omics data—we acknowledge that these are not the only challenges in this research area. Instead, they represent prominent examples that are critical to the context of cancer subtype classification using feature selection techniques. Additionally, we have explicitly included a limitation of our review, which focuses primarily on structured/tabular data and does not comprehensively address multimodal datasets. Future research and reviews should indeed explore other challenges, such as scalability, interpretability, and computational efficiency, in more detail. We appreciate your observation, which helps highlight the need for further discussion and investigation in this field.
- Figure 3 is not well described in section 5.
Answer: We thank the Reviewer for the careful observations. After careful consideration, we have decided to remove Figure 3 from the manuscript to streamline the presentation and ensure clarity. The relevant content has been appropriately incorporated into the text to maintain the flow and comprehensiveness of the discussion. We appreciate your observation, which helped us improve the quality of the manuscript.
- Do the authors consider the scope of the review to be limited to structured/tabular data? I think yes. So, my recommendation is to make this clear in the title and study objective.
Answer: We thank the Reviewer for the careful observation. The review primarily focuses on feature selection techniques applied to structured/tabular data, as this data type is commonly used in cancer subtype classification studies. While we acknowledge the importance of clarifying the scope, we believe that the current title adequately reflects the overall theme and objective of the review.
- I think the authors did not consider multimodality (multimodal datasets with structured, semi-structured, and unstructured data, such as time series, images texts, etc) in the review. However, there is a growing number of studies in this area, hence feature selection techniques should consider multimodal approaches. What do the authors think? See the following papers:
https://www.frontiersin.org/journals/artificial-intelligence/articles/10.3389/frai.2024.1408843/full
https://www.sciencedirect.com/science/article/pii/S2772941923000212
https://www.nature.com/articles/s43018-022-00388-9
Answer: We thank the Reviewer very much for raising this important point about multimodality in feature selection. We agree that integrating structured, semi-structured, and unstructured data, such as time series, images, and text, represents a growing and impactful research direction in cancer subtype classification and other biomedical applications. The references you provided further emphasize the relevance of this emerging area. While our review primarily discusses feature selection techniques within a specific context, we recognize the increasing importance of multimodal approaches. In the revised Manuscript, we have noted this as a limitation and have highlighted the need for future work to address multimodal data integration in feature selection methodologies. We appreciate your insightful comments and the references, which we believe will enrich future research in this field.

Round 2
Reviewer 2 Report
Comments and Suggestions for Authors
The authors improved the manuscript, and addressed most of my concerns. However, some comments remain:
Comment 4: The manuscript would benefit from examples for some explanations. For example, consider "This not only enhances the performance of classification algorithms but also aids in interpreting the underlying biological significance of selected features"; how does it enhance the performance and aid in the model interpretation? An example could enrich the explanation.
Answer: We thank the Reviewer for the constructive suggestions. We added an example in the revised Manuscript:
For instance, study presents a machine learning framework that integrates somatic mutation and gene expression data to classify breast cancer subtypes. The authors employ feature selection techniques to identify key genes associated with specific subtypes. These selected features improved model accuracy by reducing noise and overfitting while also providing biologically meaningful insights into the molecular mechanisms of breast cancer [43].
My reply: The authors considered only the example I provided. However, my comment was general, not focused on a specific part of the text. This comment applies to the whole manuscript.
Comment 12: As this is not a systematic review, how did you select the 'studies reviewed'?
Answer: We thank the Reviewer for the strict review of the manuscript. We have modified the sentence in the revised Manuscript:
As summarized in Table 5, the studies provide information on the cancer subtypes, the feature selection methods employed, the data types used, and the key biomarkers identified.
My reply: The authors did not answer my question. The answer should be clear in the manuscript.
Comment 15: In section 4. Challenges and Limitations, which are challenges and which are limitations? Is there any difference?
Answer: We thank the Reviewer for the careful review. In Section 4 of the manuscript, we have used the terms interchangeably to discuss both the inherent difficulties (challenges) in cancer subtype classification and the shortcomings (limitations) of existing feature selection techniques. For example:
Challenges: These refer to broader, systemic issues, such as the complexity and heterogeneity of cancer, the high dimensionality and sparsity of omics data, and the integration of multi-omics datasets. These represent unresolved problems that require innovative solutions.
Limitations: These are more specific to the methodologies used, such as the computational inefficiencies of wrapper methods, the instability of traditional feature selection methods, and the absence of standardized best practices.
My reply: This is not clear in the text at all. If the authors are using the terms interchangeably, the terms have the same meaning. So, one only should be used. If they have different meanings (as the authors tried to explain in their answer), the difference should be clear in the manuscript. In addition, the authors added a new limitation for the review, not a limitation of the research area. This makes the text even more confusing.
Author Response
We thank the Reviewer very much for the careful review and the valuable comments and suggestions. We futher modified the Manuscript accordingly.
Answer for Comment 4: We thank the Reviewer for the valuable comments, and we have revised the Manuscript.
Answer for Comment 12:
We thank the Reviewer very much for the insightful comments. The studies included were selected based on their relevance to the topic of feature selection in cancer subtype classification. The inclusion criteria focused on:
Studies published in peer-reviewed journals.
Research that explicitly applied feature selection techniques in the context of cancer subtype classification.
Studies covering diverse data types, such as genomic, transcriptomic, or multi-omics data.
The literature was primarily identified through searches in databases such as PubMed and Web of Science using keywords including 'feature selection' and 'cancer subtypes'. Articles were further screened to ensure relevance and quality based on the abstracts and full-text reviews. No specific time range was applied, but priority was given to recent advancements and high-impact studies. Additionally, reference lists of selected articles were manually checked to identify other relevant works.
Answer for Comment 15: We thank the Reviewer for this careful comment. We have clarified the distinction between "challenges" and "limitations" in Section 4 in the revised Manuscript. Specifically, challenges refer to broader, systemic issues inherent to cancer subtype classification, such as the complexity and heterogeneity of cancer, the high dimensionality and sparsity of omics data, and the integration of multi-omics datasets, representing unresolved problems that require innovative solutions. Limitations, on the other hand, are more specific shortcomings related to the methodologies used, such as the computational inefficiencies of wrapper methods, the instability of traditional feature selection techniques, and the absence of standardized best practices. Additionally, we have ensured that the discussion of the limitations of this review (e.g., its primary focus on structured/tabular data) is clearly separated from the broader challenges in the research field. We hope this revision makes the manuscript clearer and more precise.